# Coastal Flooding and Inundation and Inland Flooding due to Downstream Blocking

**Leonard J. Pietrafesa [1,2,\*], Hongyuan Zhang [1], Shaowu Bao [1], Paul T. Gayes [1] and Jason O. Hallstrom [3,4]**

[1]   Burroughs and Chapin Center for Marine and Wetland Studies, Coastal Carolina University,
     Conway, SC 29528, USA; hzhang@coastal.edu (H.Z.); sbao@coastal.edu (S.B.); ptgayes@coastal.edu (P.T.G.)

[2]   Department of Marine, Earth and Atmospheric Sciences (emeritus), North Carolina State University,
     Raleigh, NC 27695, USA

[3]   Department of Coastal and Marine Systems Science, Coastal Carolina University, Conway, SC 29528, USA;
     jhallstrom@fau.edu

[4]   Institute for Sensing and Embedded Network Engineering, Florida Atlantic University,
     Boca Raton, FL 33461, USA

\*   Correspondence: len_pietrafesa@ncsu.edu

**Abstract:** Extreme atmospheric wind and precipitation events have created extensive multiscale coastal, inland, and upland flooding in United States (U.S.) coastal states over recent decades, some of which takes days to hours to develop, while others can take only several tens of minutes and inundate a large area within a short period of time, thus being laterally explosive. However, their existence has not yet been fully recognized, and the fluid dynamics and the wide spectrum of spatial and temporal scales of these types of events are not yet well understood nor have they been mathematically modeled. If present-day outlooks of more frequent and intense precipitation events in the future are accurate, these coastal, inland and upland flood events, such as those due to Hurricanes Joaquin (2015), Matthew (2016), Harvey (2017) and Irma (2017), will continue to increase in the future. However, the question arises as to whether there has been a well-documented example of this kind of coastal, inland and upland flooding in the past? In addition, if so, are any lessons learned for the future? The short answer is "no". Fortunately, there are data from a pair of events, several decades ago—Hurricanes Dennis and Floyd in 1999—that we can turn to for guidance in how the nonlinear, multiscale fluid physics of these types of compound hazard events manifested in the past and what they portend for the future. It is of note that fifty-six lives were lost in coastal North Carolina alone from this pair of storms. In this study, the 1999 rapid coastal and inland flooding event attributed to those two consecutive hurricanes is documented and the series of physical processes and their mechanisms are analyzed. A diagnostic assessment using data and numerical models reveals the physical mechanisms of downstream blocking that occurred.

**Keywords:** downstream blocking; compound flooding; coastal storm surge and inundation; explosive lateral flooding; hurricane inland and upland flooding

## 1. Introduction

Compound hazards are those events that occur simultaneously or successively whose combination and interaction with underlying conditions amplify the hazardous impacts from individual events [1], creating storm surge and thus seawater inundation. The storm's heavy rainfall, on the other hand, causes surface runoff, sub-surface flow and river flooding. These two flooding processes can have dependence [2] and complex interactions. A higher downstream coastal water level changes the

river's downstream boundary conditions, and thus affects upstream river flow dynamics and inland freshwater flooding [3–7]. Simultaneously, the flow of river flow into the ocean can affect coastal sea level changes [8], which, in turn, can act as a feedback effect to further impact the river flooding. Jay et al. [9] and Guo et al. [10] analyzed and documented the relative importance of tides and river flows at different locations along the river channel to coastline at several floodplain wetlands.

An example of a compound hazard occurred in 1999. From late August to early September in 1999, Hurricanes Dennis and Floyd passed along and across the eastern coastal region of North Carolina (NC), and together deposited about 1000 mm of precipitation. The pair created extreme coastal, inland and upland flooding. The combined effects of the hurricanes resulted in massive property damage and led to 74 (56 in NC) human fatalities, due principally to the ensuing flooding. The net cost of the damage ascribed to the flood event was in excess of $6.5B in 1999 and 10.166B in 2019 dollars. The flood event extended from the coast to New Bern NC and Washington NC, well inland and upland. At the time, the flooding inland and upland were not associated with the downstream blocking of the Neuse and Tar-Pamlico Rivers, but rather directly with Floyd's 600 mm of rainfall.

Hildebrand [11] found that from 1887 up to 1999, NC had experienced 83 named tropical storms and 31 hurricanes, but none had resulted in the massive flooding associated with these two 1999 events. However, Pietrafesa et al. [12] further documented the revelation that flooding, instead of winds, was responsible for nominally 65% of hurricane-related property damage in NC. More recently, heavy precipitation events such as hurricanes Joaquin (635 mm rainfall) and Matthew (457 mm rainfall) in South Carolina (SC) in 2015 and 2016, respectively, hurricane Harvey in 2017 in Texas (1828 mm rainfall), and hurricane Florence in 2018 (914 mm rainfall) in NC and SC, are examples of these kinds of heavy precipitation events. Hurricane Matthew resulted in $10.3B in damage in 2016 ($10.92 in 2019) dollars in SC alone. Rivers inland in SC crested to unprecedented levels of 5–6 m over mean water levels (NCEI), such as the Waccamaw River at Freeland and the Congaree River in Columbia, well up to the foothills of the Appalachian Mountains. Following Florence (2018), the Waccamaw River crested at 0.9 m above the level reached during the passage of Matthew (2016). There are wide ranges of spatial and temporal physics scales and thus in reported impacts (p.c. from M. McClam of the South Carolina State Guard), from several kilometers to several tens of kilometers downstream, and hours to days downstream to tens to many hundreds of kilometers upstream and days to weeks and back to tens of minutes upstream. To reduce the risks from these kinds of seemingly hidden and then often explosive flooding events in the future, we need to understand the fluid mechanics of these events that are temporally extensive and spatially massive, often transitioning to short period, laterally explosive from inland to upland and from the coasts to the mountains.

This study discusses the 1999 Dennis-Floyd event and the nonlinear fluid physics that ensued, and presents this case as a precursor to future events in-kind. First proposed by Pietrafesa and Dickey at an Eastern Carolina University (ECU) Hurricane Flood Workshop, in the study reported on below, it is found that in 1999, while Hurricane Floyd was attributed solely for the inland flood damage [13], Hurricane Dennis actually set the stage for the massive inland and upland flooding by changing the downstream boundary conditions. This study documents the events that preceded, were present during, and followed the passages of Dennis and Floyd and offers the possibility of an improved model prediction scheme for inland and upland flooding in coastal states. Moreover, the need to properly initialize the water levels in prognostic numerical models of incoming heavy precipitation events is suggested as both proof of concept and as a warning for the future. The Dennis-Floyd scenario is described in the section to follow, because of the comprehensive data set that is available to study that combined event. A numerical modeling scheme is envisioned and should be developed and employed in the future, if forecasts of coastal, inland and upland flooding are to improve from present-day mathematical architectures.

## 2. Data and Study Area

The study area, which encompasses eastern NC, and the points of in situ observations, are presented in Figure 1. Data used in this study (cf. Figure 1) include time series of atmospheric winds, precipitation, water levels and water currents. Atmospheric data is from the National Weather Service (NWS) first order stations. River discharge and water level data for Little Washington NC are from the U.S. Geological Survey (USGS). Open ocean coastal sea level and sound-side water level is from the National Ocean Service (NOS). Wind data time series are from the NC Coastal-Marine Automated Network (C-MAN station) located along the coast downstream from Ocracoke Inlet where the + sign is shown, and the Kinston, NC Airport and precipitation data are from the National Weather service (NWS). Sea surface temperature (SST) data are from NOAA's polar orbiting satellite. Sea surface and cloud color data are from the NASA SeaWifs and Infrared Imager satellites. All data, including the hurricane tracks, are available from the National Center for Environmental Information (NCEI) at: https://www.ncei.noaa.gov/.

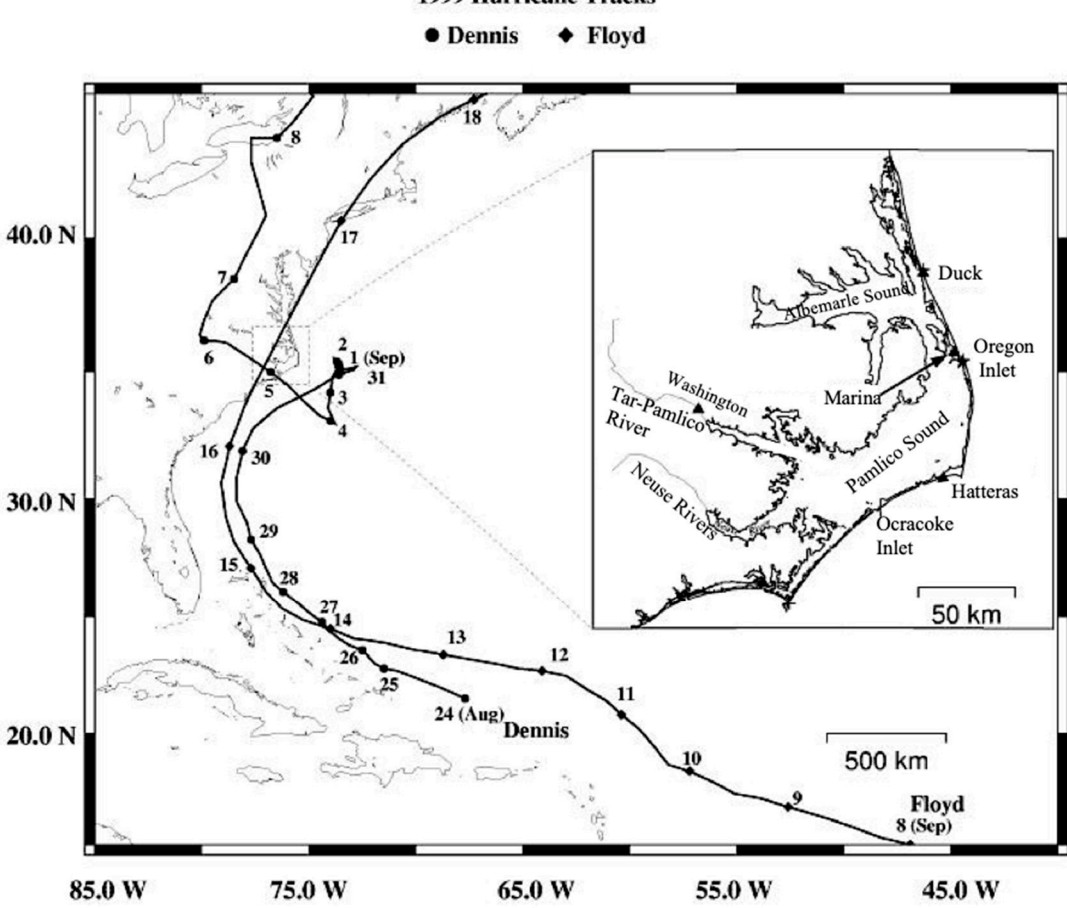

**Figure 1.** Tracks of 1999 hurricanes Dennis and Floyd. The insert is the eastern NC study area. Triangles represent NWS and USGS data collection sites, and stars represent NOS data collection sites. Data are provided by the NCEI: https://www.ncei.noaa.gov/.

## 3. Analyses

In Figure 1, we see that Hurricane Dennis entered the region of the NC coast on 30 August and became stationary for 6.5 days off the NC coast east of Cape Hatteras, finally leaving the area on 06 September.

In Figure 2a,e, we see that when the winds blew from the north and northeast, on the western or shoreward side of Dennis' eye, water levels rose within several hours on the open ocean side of the

coast at the Duck, NC. The water level (tide gage) data has been low-pass filtered using a 40-hour half power point Lanczos-Cosine filter [14]. Within the sound system, water levels fell in the northeastern end or upper Pamlico Sound, and rose in the upper Tar-Pamlico River, all within three hours. On August 30, water levels rose at Duck by 98 cm, rose at Washington by 21 cm and fell at the Marina by 22 cm. What this indicates is the quick response time of the entire Pamlico Sound to an axial wind, consistent with [15], who reported that water levels respond fully to strong along-sound axial winds within 2 h and 45 min. Thus, water levels in the southwest end of Pamlico Sound set up or rose, and water levels in the northeast end of the sound set down or dropped. The water level at the Marina was out of phase with the water levels at both Duck and Washington over the entire period extending from August 25 to September 20. The rises and falls (Figure 2a–c) and the subsequent differences (Figure 2d) coupled tightly to the NE/SW component (Figure 2e) of the total wind-field (not shown). The difference in water levels was most dramatic from August 30 to September 06, which is coincident with the presence and eventual passage of Dennis. This also had the effect of driving coastal waters towards the three inlets, Oregon, Hatteras and Ocracoke and of driving waters away from them on the Pamlico Sound sides of the inlets. This effected a double suction of water through the inlets, convergence on the outside and divergence on the inside, a process previously reported on by [14]. We show below that this set up in the SW corner had the effect of blocking the flows at the mouths of both the Tar-Pamlico and the Neuse Rivers, so water levels had to rise upstream consequently, filling the water basins to near capacity.

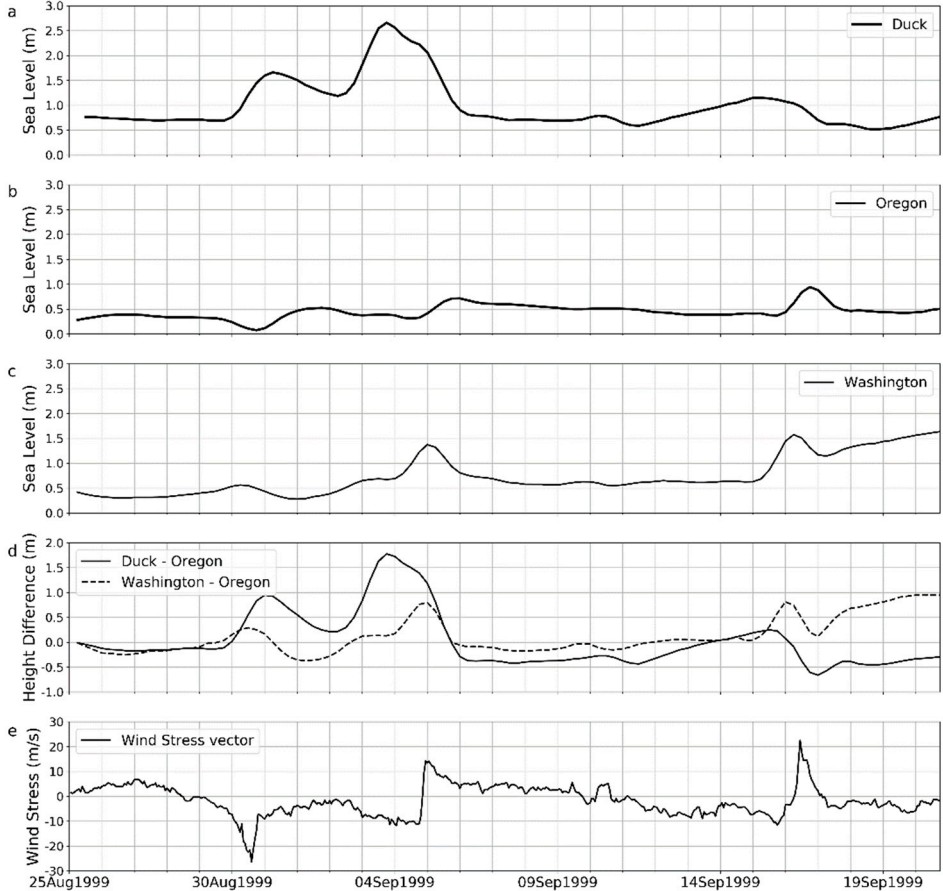

**Figure 2.** Time series of water level at Duck, NC (**a**), Oregon Inlet (**b**), Washington, NC (**c**), and their difference (**d**). The alongshore wind at Cape Lookout is shown in (**e**) with the positive sign indicating northeastward wind and the negative sign southwestward. Time period is from 08/25 to 09/20, 1999, which encompassed Hurricanes Dennis and Floyd.

Hurricane Dennis then wobbled somewhat off the coast, so at any location the relative wind field changed in intensity with time, while about 280 mm of rain was deposited over the coastal region and the water levels fluctuated. On September 05, Duck water levels peaked at 196 cm higher than they were prior to Dennis' incursion and Washington water levels reached 109 cm higher than they were prior to Dennis' arrival. Inshore Marina water levels were 177 cm below coastal water levels and 85 cm below Washington water levels. Then, as Dennis moved across and finally departed the state on September 06, offshore coastal and upstream river water levels began to return to their prior state.

### 3.1. Water Transport through the Inlets

The Nichols and Pietrafesa [14] study showed that for periods in excess of a day, the axial flow through Oregon Inlet and sea level slope, are tightly coupled in a 43 cm/sec/meter relationship. Using this stable transform function, we can compute the volumetric flux of water through the inlet. We note that the three inlets from the coastal ocean to Pamlico Sound, Oregon, Hatteras and Ocracoke, are but several km in width, and thus are very spatially narrow. Using 7000 m² [14] as the nominal cross section of Oregon Inlet, and an amount in kind for Ocracoke and Hatteras Inlets taken together, we compute the time series of volumetric flux, the cumulative flux of water either in or out of Pamlico Sound is shown in Figure 3. Here we see that following the onset of Dennis, coastal waters began to flood Pamlico Sound via Oregon, Ocracoke and Hatteras Inlets on 30 August and continued doing so until 06 September. During this period, the flux of shelf water into the sound occurred at non-tidal speeds occasionally reaching nearly 1.5 m/s, and the added amount of water that entered Pamlico Sound via Oregon Inlet alone reached the volumetric value of $1.4 \times 10^9$ m³. Ocracoke and Hatteras added another $1.25 \times 10^8$ m³. The salt concentrations of the water masses entering the sound reached 30 parts/1000 or ppt (not shown).

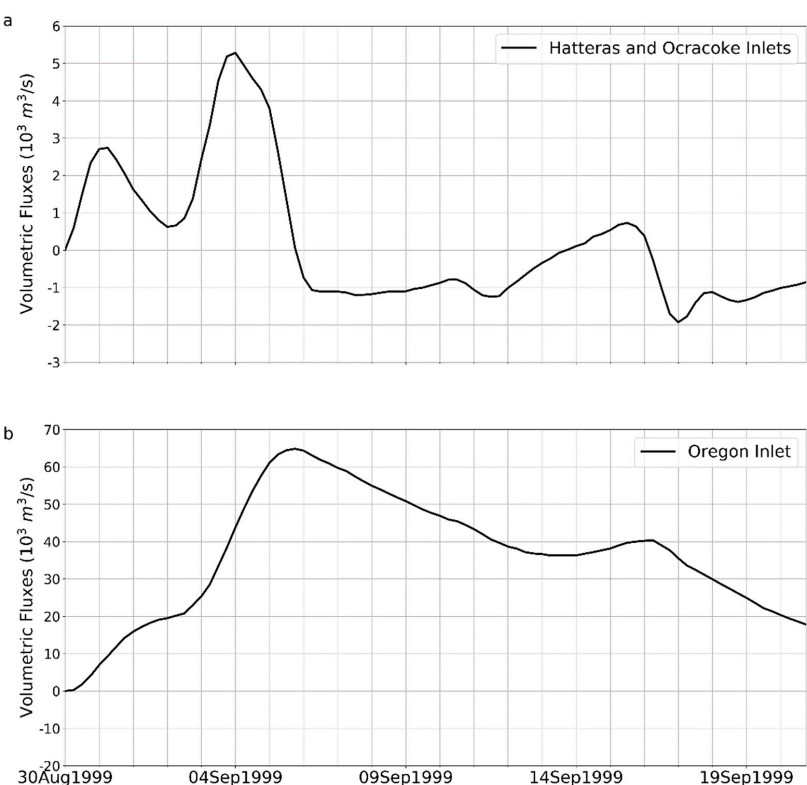

**Figure 3.** Volumetric Fluxes of water during the passages of Dennis and Floyd (30 August–20 September) through: Hatteras and Ocracoke Inlets (**a**) and Oregon Inlet (**b**). Positive (Negative) is into (out of) Pamlico Sound.

Using the NOS water level records and the estimates of the size of the sound proper [15], we calculate the amount of water that was present in Pamlico Sound at the end of August, prior to the arrival of Dennis, was approximately $1.86 \times 10^9$ m$^3$. Thus, the amount that entered the Sound over the 6.5-day period when Dennis was present increased the total amount of water in Pamlico Sound by 75% to $3.26 \times 10^9$ m$^3$ of water. This additional water flooded the low-lying perimeter of Pamlico Sound.

Following Dennis' departure, the Pamlico Sound system began to drain, through all three barrier island inlets. The total volume of water decreased by about $0.8 \times 10^9$ m$^3$, but as the M2 tide and weak axial pressure gradient flows were present to drain waters through the inlets (acting as outlets) the sound still retained $2.46 \times 10^9$ m$^3$ by September 13. However, on September 14 the waters began to rise again as Floyd approached, bringing more precipitation (cf. Figure 4a,b and Figure 2b), along with winds favorable for additional incursions of coastal waters into the sound. By September 16, the additional amount of $0.1 \times 10^9$ m$^3$ of water added to the system, reached $2.56 \times 10^9$ m$^3$ by September 16. This water level was still 38% higher than prior to the arrival of either Dennis or Floyd. Subsequently, the water level gradually fell when the system drained out of the inlets. By September 21, the volume of water in Pamlico Sound proper had declined to $2.25 \times 10^9$ m$^3$ or 22% more water than was present on 30 August. As can be seen in Figure 1, Hurricane Floyd was present from September 15–17.

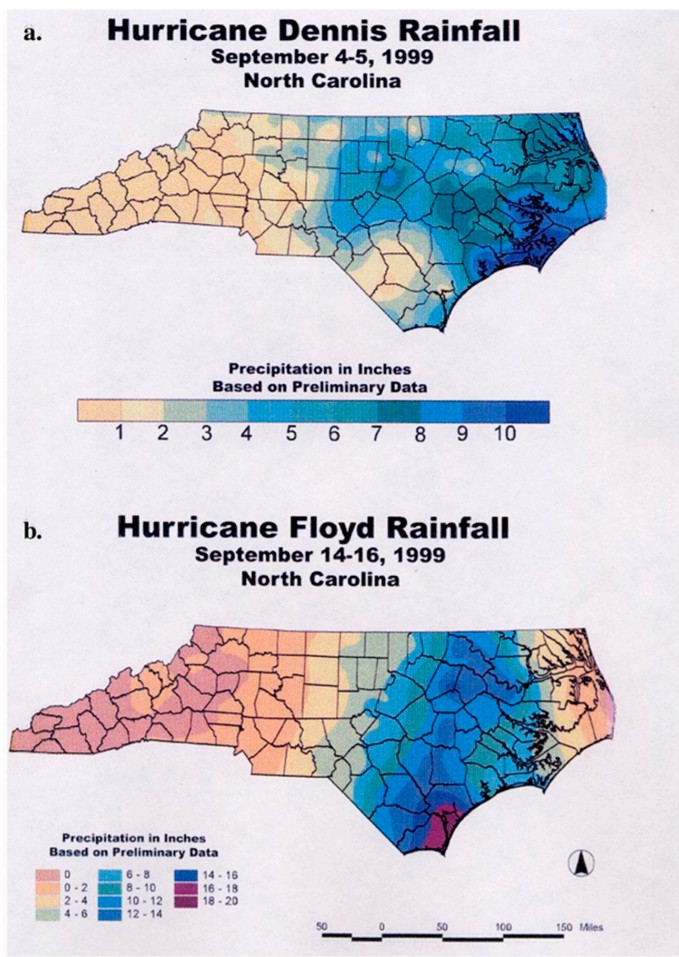

**Figure 4.** Rainfall bands across NC associated with (**a**) Dennis and (**b**) Floyd in 1999.

## 3.2. The Water Level Blocking Effect and Subsequent Flooding

The data and the imagery presented show that when Hurricane Dennis sat off the NC coast, it created conditions favorable for the flooding of the Pamlico Sound with coastal waters for 6.5 days. The enormous amount of highly saline coastal water that entered the system added 75% more water to what was already present in Pamlico Sound. Following Dennis' departure, all three sound inlets began

to drain using the increased pressure gradient forces from inside to outside the sound along the three narrow inlet axes, as the driving forces. The time series of water levels at the Marina, inside the sound and near Oregon Inlet, and the upper Tar-Pamlico River (Figure 2b,c) and the differences between the two (Figure 2d) offer further evidence for this scenario, which indicates that Pamlico Sound was filled to capacity and thus blocked the mouths of the river-estuary tributaries, specifically the Neuse and Tar-Pamlico Rivers. Then while the system was still backed up, or rather was in a storage mode, and while the region's soils and vegetation were still highly saturated, along came Hurricane Floyd, dropping a then historic record amount of rain over a large swath of NC's coastal and eastern middle interior, inland and upland (Figure 4). The waters in the sound proper were not able to drain to the sea quickly enough as there were no driving forces other than inlet, along river axes pressure gradients. Subsequently the rivers swelled over their banks, drowning the coastal plain and the inshore areas laterally. Moreover, the lateral movement occurred explosively in tens of minutes. The September 23 NASA SeaWifs image of the sound region (Figure 5) strongly suggests that the highly turbid river waters had not yet reached Pamlico Sound proper, though they are clearly present in the rivers.

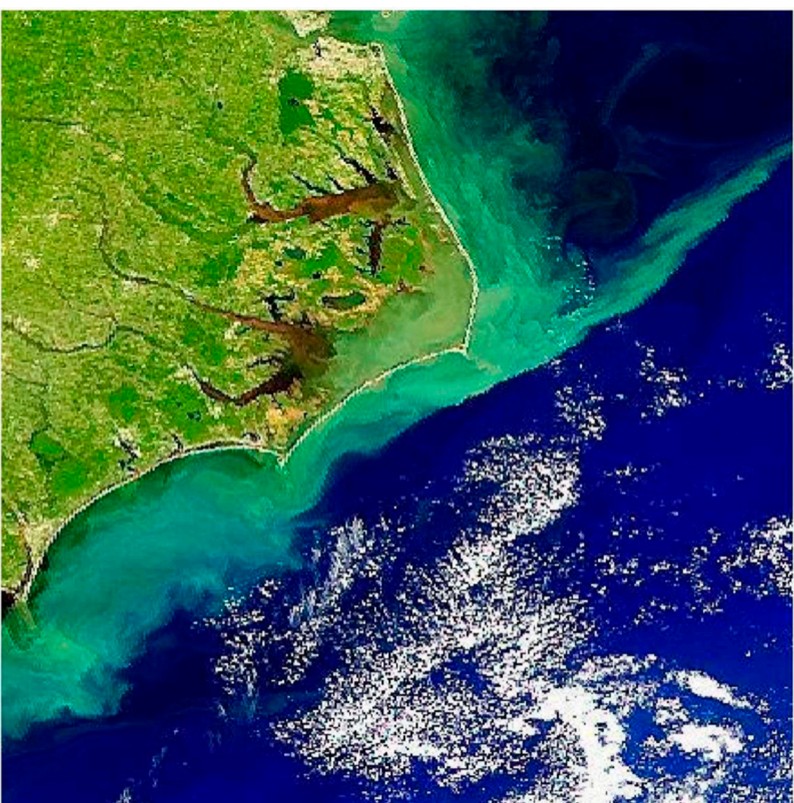

**Figure 5.** NASA September 23 SeaWifs satellite image showing surface color following the passage of Hurricane Floyd. Notice the waters in the sound are visually different from those in the Neuse and Tar-Pamlico Rivers.

To assess this possible scenario, we look at the actual daily time series of stream-flow data from both the Neuse and the Tar-Pamlico Rivers. In Figure 6a, following August 30, the flows in both rivers actually dropped until 04 September 04. In Figure 6b, when the flux began to accelerate until September 16–17 when the flows rapidly intensified. By September 19, the total discharge from the two rivers reached the amount of water that entered the sound from offshore during the oceanic flood caused by Dennis. Therefore, Dennis and Floyd acted in concert to cause the extensive flooding. The flooding was so extensive that it reached Greenville, Washington, Kinston and New Bern, NC, cities (not shown) all of the order of tens and hundreds of kilometers inland and upland from Pamlico Sound and the Outer Banks, NC.

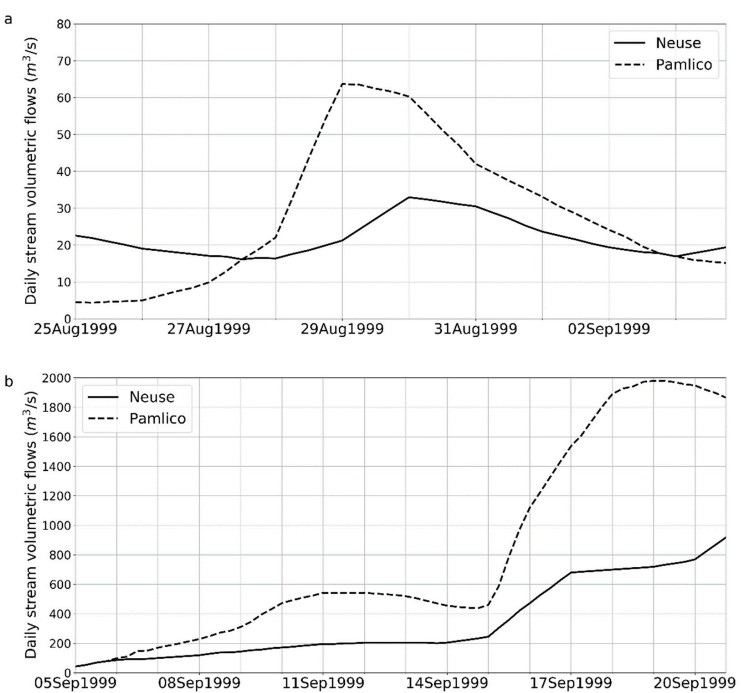

**Figure 6.** Daily stream volumetric flows in the Tar-Pamlico and Neuse Rivers for the period of (**a**) August 25 to September 04 and (**b**) September 04 to September 20, 1999.

## 4. Numerical Model Testing

To understand the mechanism of the hydrological process caused by two continuous hurricanes, an idealized one-dimensional hydrological model based on the Saint-Venant (S-V) equations (Equations (1) and (2)), is designed. While a state-of-the science sophisticated three-dimensional model could be applied to reveal the total physics of the compound flooding phenomena in the eastern NC setting, we employ the S-V model to provide simplified, yet revealing foundational physics to the compound flooding phenomena.

$$\frac{\partial A}{\partial t} + \frac{\partial Q}{\partial x} = q_{lat} \tag{1}$$

$$\frac{\partial Q}{\partial t} + \frac{\partial\left(Q^2/A\right)}{\partial x} + Ag\frac{\partial h}{\partial x} + AgS_f = 0 \tag{2}$$

$A$ is the flow area of cross-section. $Q$ is flow rate. $q_{lat}$ is lateral inflow rate into the channel from rainfall and surface runoff. $h$ is water surface elevation. $S_f$ is friction slope, defined as $S_f = (Q/K)^2$ where K is conveyance from Manning's equation, defined as $K = \frac{1}{n}AR^{2/3}$, where n is Manning's roughness coefficient, R is hydraulic radius R=A/P, P is wetted perimeter. The first two terms of the momentum equation (Equation 2) were ignored; therefore, the momentum equation was simplified as a diffusion wave equation. This implementation of the St. Venant equations is similar to the one used in the WRF-hydro model [16].

$$Ag\frac{\partial h}{\partial x} + AgS_f = 0 \tag{3}$$

The idealized 1-D model employed a 10 km linear river channel with a bed slope of 0.0002. The initial river water depth is set at 1 m. It is assumed that the vertical dimension of the river channel is effectively infinite [16]. Two downstream sea levels were used: 1 m (Control Experiment) as the normal condition, which is the same as the normal river surface height, and 1.5 m (Δ-Sea-Level Experiment), as the condition after the hurricane-induced storm surge. An idealized rainfall amount was added to the river channel. Here we add a lateral discharge with value of 30 m³/s to model one example of the

rainfall process. This rainfall has a duration of 30 min. The rainfall line shown in Figure 7 illustrates where the rainfall occurs (3 to 4 km from the river mouth). Results of this model are shown in Figure 7. About half hour after the rainfall was deposited onto and into the river channel, the surface water height in the Δ-Sea-Level Experiment was higher than the Control Experiment up to 2 km from the downstream boundary, and its discharge was slightly less than the Control Experiment 3 h after the rainfall, the higher surface water level intrudes further toward inland reaching up to 4 km from the downstream boundary. The surface water slope in the Δ-Sea-Level Experiment is lower than the Control Experiment, resulting in slower water flow speed (*V*). However, the higher water surface in Δ-Sea-Level also created a large river cross-section area (*A*); therefore, the simulated streamflow discharges ($Q = V \times A$) in the Control Experiment and the Δ-Sea-Level Experiment converge after 3 h. Note in this idealized experiment the vertical dimension of the river channel is assumed to be infinite, meaning the increased water can be stored in the river channel. However, in real-world conditions, this increased water will move laterally and become flood waters.

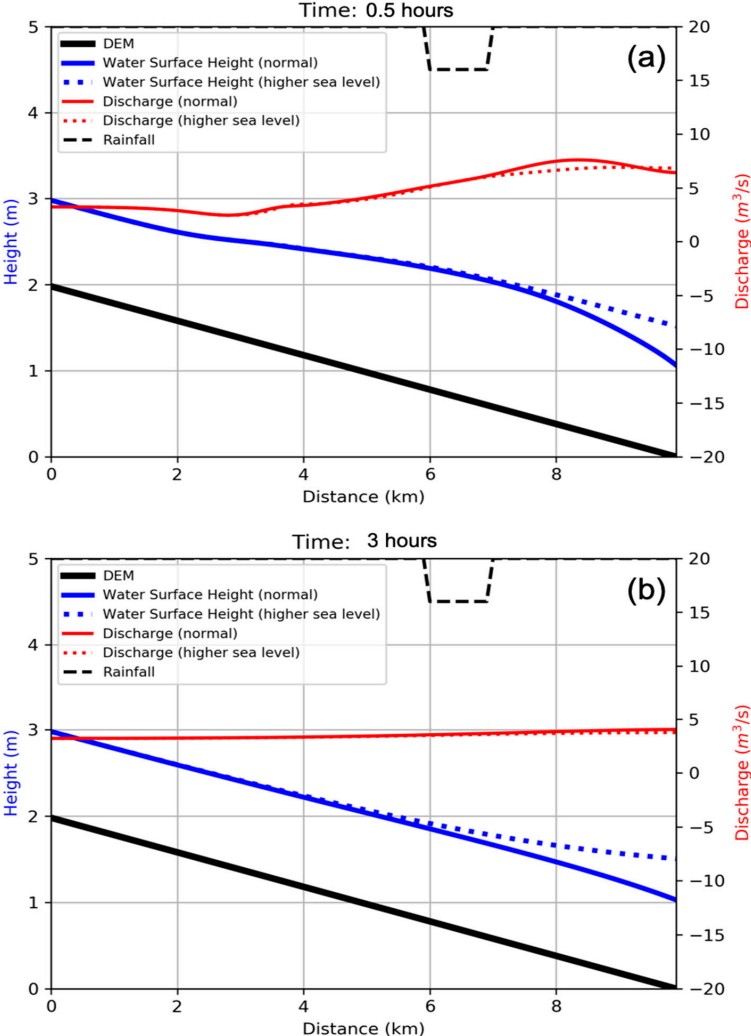

**Figure 7.** One-dimensional idealized river channel simulation results: (**a**) 30 min after the rain event; and (**b**) 3 h after the rainfall was added to the river channel. The black line is the constant riverbed. The blue lines are the water surface height in the river channel. The red lines denote the streamflow in m³/sec. The bold solid lines show the results from the simulation with normal sea level as the downstream boundary condition, and the dashed lines are those with increased sea level as the initial condition.

Based upon the 1-D idealized numerical experiment, the Princeton Ocean Model (POM), developed by [17] was also used to simulate the above hypothesis. The model incorporates the Mellor–Yamada turbulence scheme, has a free surface to handle tides, sigma vertical coordinates (i.e., terrain-following) to handle complex topographies and shallow regions, a curvilinear grid to better handle coastlines. We employed the POM community model to emulate Pamlico Sound with lateral topography similar to eastern NC and applied it multiple times and by raising the initial water level at the mouths of the two rivers in sequential model runs. In Figure 8a, the initial water level affects the subsequent Hurricane Floyd induced storm surge when the storm surge rises linearly and then becomes highly nonlinear. In addition, in Figure 8b, the initial water level played an even larger role in the lateral flooding of land, with the lateral flooding increases at an increasing rate shown as a nonlinear power law. At zero initial water level, the lateral flooding encompassed about 250 $km^2$. However, at 2 m above zero, the POM computed lateral flooding rises nonlinearly to 1700 $km^2$, a 7-fold physics-based nonlinear increase.

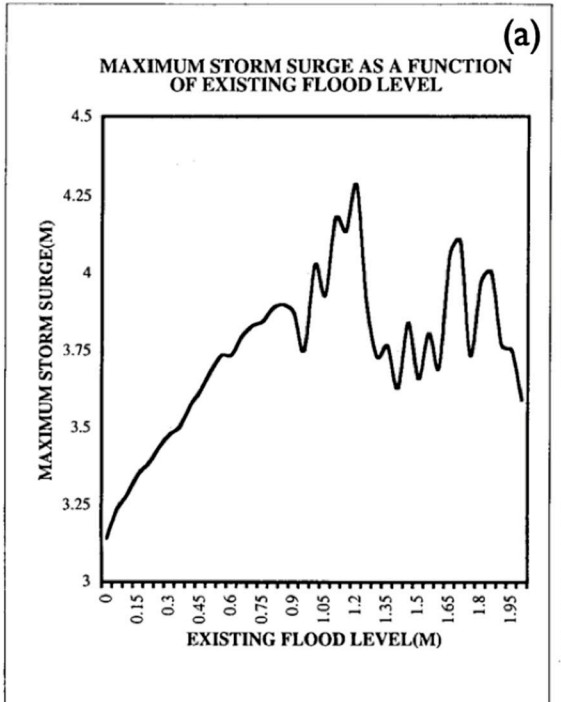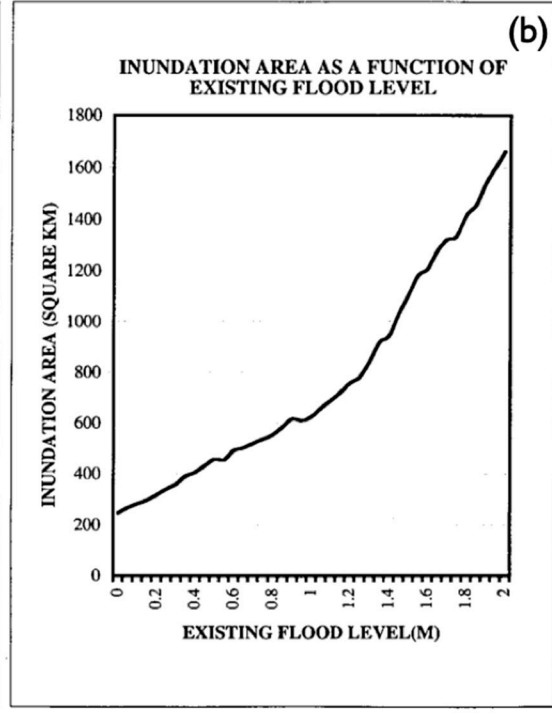

**Figure 8.** Raising the initial water levels (referred to as the "Existing Flood Level" along the horizontal axis) incrementally for the Pamlico Sound System and then employing the POM model to compute the: (**a**) resulting storm surge; and (**b**) the lateral inundation of the domain.

This simple rendering suggests that when coastal rivers, that is, coastal watersheds, can no longer rise upwards and their carrying capacity to relieve themselves downstream is compromised, additional amounts of rainwater and land runoff will initiate an explosive lateral flooding event. The word 'explosive' is employed because many of the 56 NC casualties occurred on the highways, suggesting that the roads suddenly, within tens of minutes, were inundated with several feet of water and the cars were washed off of the roads and went down the embankments. This phenomenon must be addressed in future diagnostic and prognostic numerical modeling architectures. In fact, the National Water Model (NWM) presently used by the NOAA does not connect to coastal waterbodies on any U.S. coastline, including the Great Lakes. Furthermore, groundwater sources of rainwater can reappear days to weeks following an event or series of events and amplify the unforeseen explosive floods, well upstream, both inland and upland. River water levels may be rising because water is flowing upstream or simply rising in place. Coastal watersheds can transition into storage modes due to downstream blocking; a non-local forcing effect.

## 5. Discussion, Conclusions and Recommendations

In this study, a 1999 rapid coastal and inland flooding event attributed to two consecutive hurricanes Dennis and Floyd is documented. In summary, the series of physical processes and their mechanisms are the following. (1) Dennis delivered 381 mm of precipitation. (2) Dennis' translational speed slowed to near zero and Dennis hovered off the NC coast for 6.5 days. (3) Dennis' winds mechanically drove coastal waters towards the coast, and within 8 h of onset built up a wall of water along the offshore side of the NC Outer banks coast. These same winds simultaneously drove inshore sound waters from the northeast end of Pamlico Sound towards the southwest end within 3 h. (4) Pamlico Sound was flooded by relatively salty coastal ocean waters. (5) The amount of ocean water which entered the sound system during Dennis' presence was equivalent to 75% of the amount of water already present in Pamlico Sound. (6) The excessive amount of water in Pamlico Sound blocked the flows from the Neuse and Tar-Pamlico rivers, causing the two rivers to go into relative storage modes and thus backed waters up towards the heads of the rivers, filling the watersheds to near capacity, thereby creating the conditions for explosive lateral flooding. (7) Following Dennis' departure, the waters in the sound began to discharge through the three barrier island inlets but before the waters could drain, along came another wet hurricane. (8) Hurricane Floyd deposited a then-record 609 mm rainfall onto already saturated soils. (9) When Floyd arrived, river waters were still blocked at their mouths, and river waters expanded laterally over their banks, thereby flooding the watersheds to record levels many tens to hundreds of kilometers inland and upland. (10) Following Floyd's departure, the entire sound system began to drain through its three barrier island inlets, now functioning as outlets, and continued doing so for several months.

Unfortunately, although the phenomenon of compound flooding has been recognized qualitatively on the conceptual level, the skills of quantitatively predicting such events using numerical models remain poor. The deficiency has been due to the following two reasons. First, traditionally compound events have been treated via a "top-down" perspective, which typically only considers one event and its hazard at a time, potentially leading to an underestimation of risk, as the processes that cause extreme events often interact and are spatially and/or temporally dependent [18]. Such a perspective has led to the fact that in the U.S., flood hazard assessment practices are typically based on univariate methods. Thus, hydrology and oceanography modelers often concentrate only on their own respective domains. For example, procedures for rivers often treat oceanic contributions (e.g., storm surges) using static base flood levels, and do not consider the dynamic effects of coastal water levels. Similarly, flood hazard procedures for coastal storm surge and seawater inundation do not account for terrestrial factors such as river discharge or direct precipitation into urban areas [3,19–21]. Additionally, the National Water Model (NWM), the main tool for the National Weather Service (NWS) to forecast flood events and to issue flood warnings, currently runs on a national domain that cover a majority of the country with gaps along coastal estuaries [22]. Therefore, in order for the coastal and inland flooding attributed to events like hurricanes Dennis (1999) and Floyd (1999) to be predicted in the future, what is required is an atmospheric-oceanic-land-hydrology-hydraulic coupled model system with downstream boundary condition and initial condition properly represented.

The multiscale, multi-physics aspects of this study are considerable and highly informationally provocative. Coastal storm surges are on the order of a meter or more in the vertical and extend alongshore for coastal distances of several kilometers, with a persistence of over several hours to several days. Coastal water level spin-up times are 8 h [23,24]. These wind-driven conditions ride atop semi-diurnal and diurnal tides. Those joint conditions can then ride atop elevated coastal water levels of order kilometers from any prior events, which have left elevated water levels at the mouths of rivers, harbors and estuaries. These conditions then ride atop North Atlantic Ocean Basin seasonal, 3 to 6 months, steric adjustments of a up to 30 cm, over tens to hundreds of kilometers to annually adjusted global sea level rise. All of these are initial conditions and coastal boundary conditions. Then we demonstrated that by elevating initialized sea level at the coast, the storm surge rises but then oscillates in non-stationary and non-linear physics over the periods of hours, while the lateral

inundation expands from several kilometers to hundreds of kilometers over collapsing periods down to several tens of minutes. The nonstationary and non-linear fluid multi-scale physics are revealed in this study.

**Author Contributions:** Individual co-author contributions are: (a) Conceptualization, All; (b) (c) Methodology, All; (c) Software, L.J.P., H.Z., S.B., J.O.H.; (d) Validation, L.J.P., S.B., H.Z., P.T.G.; (e) Formal Analysis, All; (f) Investigation, All; (g) Resources, P.T.G., J.O.H., S.B.; (h) Data Curation, L.J.P., S.B., H.Z.; (i) Writing, Original Draft Preparation, L.J.P., S.B., H.Z., P.T.G.; (j) Writing, Review and Editing, H.Z., S.B., L.J.P.; (k) Visualization, S.B., H.Z., L.J.P.; (l) Supervision, L.J.P., S.B., P.T.G., J.O.H.; Project Administration, P.T.G., J.O.H., S.B.; (l) Funding acquisition, J.O.H., P.T.G., S.B.

**Funding:** This research was funded by National Science Foundation, grant number CSR 1714015 and CSR 1763294. This work is also supported in part by the Major Research Instrumentation program at the National Science Foundation under award AGS-1624068.

**Acknowledgments:** The authors wish to thank the U.S. National Science Foundation (NSF) under NSF Grants CSR 1714015 and CSR 1763294, and the Coastal Carolina University Burroughs and Chapin Center for Marine and Wetland Studies for having provided support and facilities necessary to conduct this study.

**Conflicts of Interest:** The authors declare no conflict of interest.

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
