# Peer review of "Coastal Flooding and Inundation and Inland Flooding due to Downstream Blocking"

_jmse, doi:10.3390/jmse7100336_

Round 1

Reviewer 1 Report

Pietrafesa et al Review

Summary

This paper presents some observations of water level and discharge conditions along the North Carolina coast during the passage of Hurricanes Dennis and Floyd in 1999, and highlights the importance of compound flooding and the need for a more sophisticated modeling approach to couple oceanic and riverine processes during extreme conditions. The topic of compound flood risk is a highly relevant topic that is clearly in need of more research. However, this paper merely scratches the surface for how to fill this research niche. The writing is often disjointed  (bordering on stream of consciousness in the introduction) and the figure resolution needs to be improved and sharpened up. More importantly, the manuscript, though it presents some interesting observations of a compound flood event, does not make a novel contribution to our understanding of these events, and only quickly suggests at the end of the paper a coupled modeling approach that could be a solution to the more simplistic approaches. Moreover, this paper does not demonstrate that those simplistic approaches are necessarily inferior. The abstract promises a lot more than the paper delivers- what is needed is a comprehensive evaluation of the present-day model limitations, and a robust demonstration of the superior results provided by the proposed approach. The paper is incomplete, and needs considerably more development to be considered for publication.

Specific Comments

Line 10- don’t understand  “laterally explosive”

Line 47- put this figure in 2019 dollars

Line 50- need study area figure showing these locations

Line 68- this paragraph is rather disjointed, and where are the citations for all this information?

Line 92- every place name and instrument location listed in this paragraph should be in your study area figure

Figure 1- the resolution and overall quality of this figure could be greatly improved

Figure 2- is the tide removed from this time-series?

Line 222- not sure this older and heavily idealized model contributes significantly to understanding this highly complex system

Line 259-  this is a nice finding albeit no altogether unique

Lines 273-292- nice summary

Line 317-should be labeled as Figure 9, not 1

Line 322- a coupled model is a logical next step, but this paper has not demonstrated that the older approaches are inferior

Author Response

We agree with All the Points of your Constructive Critique and have revised the manuscript accordingly.

Reviewer 2 Report

The paper is interesting even if it discusses a very old event, i.e. the inland flooding induced by two consecutive hurricanes, Dennis and Floyd, that occurred in 1999. The paper should provide a valid justification for the late description of the event, and at least be very accurate in producing all the most recent literature.

The message raised by the paper is sound. However, the concept is not always supported in a linear way. The proposed numerical simulations fail to really contribute to the objective, and similarly the discussion on the 9th figure (actually the text calls it Figure 1, but it is located beween Fig.s 8 and10) does not discusses the main concept of the paper, i.e. the hazard posed by the sequence of events. It focuses on the obvious importance of the downstream boundary condition during a river flood, a well-known issue discussed by junior school textbooks. The figure should rather show the effect of a sequence of floods.

Specific comments.

Line 39: the flow… affect coastal sea level changes. The effects are reasonably minor, I suggest to remove this comment.

Line 59. 6-6  over what? Not clear

Figure 1. Not very nice figure. The two tracks should be in color, the underlying geographic chart has a low quality. Since you discuss New Bern, may be put it into the map (Washington NC is already in the map)

Figure 2. In the lower panel, there is no location (Cape Lookout) in the legend, and the panel legends are therefore not uniform.

Figure 3. It is not clear what the graph represent, since the unit measure is m3 but the y-label says Fluxes

Lines 152-157. Not clear. Why Thus?

Section 3.2. There is no need to explain in childish words the effect of a high boundary condition on a river flow. Soap, bathtub, are all inappropriate. Shorten and make it more scientific.

Figure 6. Use m, s

Section 4. Add an eq. 3 where you describe your model. Else it looks like you use eq. 1 and 2.

Fig. 7. Not clear what is the rainfall line, what is its axis?

Lines 245-249. The description of the POM model is too short

Lines 250-271. The text is not very clear. It is not clear how the numerical investigation covers the issue raised in the motivations and objectives.

Fig at page 14. Remove or modify as explained above.

References. The literature review is largely insufficient, and not updated.  I suggest the authors to give a look at the rich research offer given by journals of MDPI, with many open access papers also in the topic of coastal flooding hazard.

Author Response

We accept All of your Comments and Suggestions and have made the appropriate changes in the revised manuscript.

Thank You.

Round 2

Reviewer 1 Report

The authors present novel observations of compound flooding during the passage of two well known hurricanes in 1999 along the North Carolina coast. They highlight the critical and complex issue of compound flooding, and how it is often neglected in practice, though this is rapidly changing and the literature cited here is incomplete. The authors have mainly paid lip service to the reviewers' suggestions, and did very little to improve the manucript over the first version. That being said, the topic is of high interest to the journal's readership and the appoach is adequate, highlighted by the unique observations. The very specific suggestion at the end of how to address compound flood risk in a numerical modeling framework could be deleted, as it really doesn't add to the thrust of the paper. Therefore, I think the authors should revisit the initial reviews to tidy up the paper and beef up the literature search a bit, paying close attention to work published in the last few years. A few examples of more recent papers are cited below. Otherwise, I think the paper is suited for publication.

Jay, D.A., et al., Tidal-Fluvial and Estuarine Processes in the Lower Columbia River: I. Along-Channel Water Level Variations, Pacific Ocean to Bonneville Dam. Estuaries and Coasts, 2015. 38(2): p. 415-433.10.1007/s12237-014-9819-0

Guo, L., et al., River-tide dynamics: Exploration of nonstationary and nonlinear tidal behavior in the Yangtze River estuary. Journal of Geophysical Research: Oceans, 2015. 120(5): p. 3499-3521.10.1002/2014jc010491

Zheng, F., et al., Modeling dependence between extreme rainfall and storm surge to estimate coastal flooding risk. Water Resources Research, 2014. 50(3): p. 2050-2071.10.1002/2013wr014616

Herdman, L.M.M., Erikson, L.H. and Barnard, P.L., 2018. Storm surge propagation and flooding in small tidal rivers during events of mixed coastal and fluvial influence. Journal of Marine Science and Engineering, Volume 6 (Issue 4), Article 158, 26 pp., https://doi.org/10.3390/jmse6040158

Author Response

In the attached text you will see that we have addressed every criticism, each suggestion and all of the edits that you recommended.

We appreciate your careful and thoughtful review of our manuscript and believe that you will find it meets all of your criteria.

Reviewer 2 Report

The paper has improved.

Author Response

We agree with Your comments and recommendations.

We have added more references and have added the paragraph below to the Discussion Section.

Thank You.
